# The Epidemiology of Bruxism in Relation to Psychological Factors

**DOI:** 10.3390/ijerph19020691

**Published:** 2022-01-08

**Authors:** Mirela Ioana Flueraşu, Ioana Corina Bocşan, Ioan-Andrei Țig, Simona Maria Iacob, Daniela Popa, Smaranda Buduru

**Affiliations:** 1Department IV, “Iuliu Hatieganu” University of Medicine and Pharmacy, 400000 Cluj Napoca, Romania; mfluerasu@yahoo.com (M.I.F.); simona72cj@yahoo.com (S.M.I.); smarandabuduru@yahoo.com (S.B.); 2Department of Pharmacology, Toxicology and Clinical Pharmacology “Iuliu Hatieganu” University of Medicine and Pharmacy, 400000 Cluj Napoca, Romania; corinabocsan@yahoo.com; 3Faculty of Medicine and Pharmacy, University of Oradea, 410087 Oradea, Romania; nelutig@gmail.com

**Keywords:** bruxism, stress, anxiety, depression, students

## Abstract

The aim of the present study was to establish the prevalence of sleep/awake bruxism among young students in Transylvania and to correlate the presence of this muscle activity with behavioral variations. This analytical, observational, cohort, cross-sectional, and prospective study involved 308 volunteers aged between 19 and 30 years of different nationalities, all students of the “Iuliu Hațieganu” University of Medicine and Pharmacy in Cluj-Napoca, Romania. Subjects were asked to complete an anonymous questionnaire which was structured in five sections. The results obtained from the questionnaires were analyzed separately for sleep bruxism and for awake bruxism. We did not find any statistically significant correlation between awake bruxism or sleep bruxism and age (*p* = 0.30 and *p* = 0.37, respectively), sex (*p* = 0.44 and *p* = 0.48, respectively), or nationality (*p* = 0.55 and *p* = 0.67, respectively). Only a high degree of stress and frustration (*p* = 0.035 and *p* = 0.020) was observed in European subjects except for the Romanians and the French, likely related to the difficulties of adapting to the language and lifestyle in Romania. Female sex was statistically significantly associated with an increased level of stress (*p* = 0.004), duty-related depression (*p* = 0.006), and duty-related anxiety (*p* = 0.003). Stress and anxiety can be favorable factors in the appearance of both types of bruxism; however, depression is associated only with awake bruxism.

## 1. Introduction

Bruxism is defined as repetitive masticatory muscle activity that is characterized by clenching or grinding of the teeth and/or by bracing or thrusting of the mandible, which is specified as either sleep bruxism or awake bruxism, depending on its circadian phenotype. Sleep bruxism is a masticatory muscle activity during sleep that is characterized as rhythmic (phasic) or nonrhythmic (tonic) and is not a movement disorder or a sleep disorder in otherwise healthy individuals. Awake bruxism is a masticatory muscle activity during wakefulness that is characterized by repetitive or sustained tooth contact and/or by bracing or thrusting of the mandible and is not a movement disorder in otherwise healthy individuals [1].

Another definition describes bruxism as a nonfunctional activity of dento-maxillary apparatus characterized by repetitive, unconscious movements, such as gnashing, rubbing or collision of teeth [2], with a high frequency in the adult population, between 5.9% and 49% [3], depending on the diagnostic mode. According to the circadian rhythm, two types of bruxism are described: sleep bruxism (SB) and awake bruxism (AB). The term awake bruxism refers to the clenching of teeth and jaws during wakefulness. Sleep bruxism defined as clenching or grinding of teeth during sleep, and it is classified as a sleep-related movement disorder that occurs as a response to micro-arousals during sleep [4].

According to the literature, the prevalence of bruxism does not vary according to nationality or race [5]. The literature could not establish with certainty a difference in frequency between the two sexes [4]. Depending on the study group and the research protocol, some authors indicated a higher prevalence of awake bruxism [6] or both types of bruxism [7] in women. The presence of bruxism can vary throughout life depending on the patient’s age, with a higher frequency in young adults and a lower frequency in the elderly [8]. Although it is considered to be of multifactorial origin, cognitive and behavioral disorders play a major role in the occurrence of this dysfunction [9,10,11]. Stress, anxiety, and depression are the psychological factors most commonly associated with the presence of bruxism [12].

Stress is a word with universal use, in physics, psychology, and sociology or in connection with the profession. Hans Selye, in 1936, introduced this term in the medical vocabulary to define “a nonspecific response of the body to any stimulus” [13]. Therefore, stress defines any extrinsic or intrinsic stimulus that determines a biological response. Stress has an important subjective component, in the sense that what is challenging, easy, or even relaxing for one person can become threatening or impossible to achieve for another [14].

In many cases, the pathophysiological complications of diseases occur as a result of stress, and subjects exposed to stress, i.e., those who work or live in a stressful environment, are more likely to develop various diseases. Psychological disorders can lead to pathological conditions, such as the following:

-physiopathological reactions: tachycardia and palpitations, migraines, vertigo, intestinal transit disorders, or fatigue;-cognitive reactions: memory disorders, attention deficit, decreased ability to concentrate, or reduced flexibility;-emotional reactions: irritability, repression of emotions, emotional instability, anxiety, or depression;-behavioral reactions: sleep disorders, inefficient time management, excessive consumption of alcohol and/or drugs, or the development of hostile, aggressive behaviors [15,16].

Anxiety is a pathological affective state characterized by psychomotor anxiety or unexplained fear, without object, or related to the possibility of imminent danger or failure. Anxiety is perceived by the subject as an emotional disorder translated into an indefinite feeling of insecurity. It describes a “normal” anxiety, which creates an optimal environment for learning and performance, and a pathological anxiety, in which the subject is so deeply marked that they can no longer control themself. The pathological form is accompanied by multiple vegetative reactions [15,16].

Depression manifests itself as a state of sadness, pessimism, general disinterest, and lack of power, as well as the desire to concentrate and communicate. It can have, as a starting point, failure, loss, a state of accentuated or prolonged stress, or certain failures or incapacities. In moderate depression, the subject shows inhibition in expression, along with a pessimistic attitude. In cases of deep depression, the facies is altered and expressionless, and the subject lacks initiative [15,16].

Anxiety and depression often begin in adolescence or early youth and worsen in adulthood [17]. Depression–anxiety comorbidity represents 25% of the general pathology. Approximately 85% of patients with depression also have an increased level of anxiety, and 90% of patients with anxiety disorder are depressed [18]. In the general population, one in five subjects experience an anxious clinical episode at least once in their lifetime [19]. An increased prevalence of anxiety and depression was observed among women compared to men only before menopause [20]. The initial symptoms are vague and nonspecific; hence, only one-quarter of cases of depression or anxiety benefit from therapeutic intervention.

Frustration is a psycho-emotional state generated by the refusal or prohibition of satisfying certain desires or requirements of instinctual origin or as a result of irrational beliefs [21].

Young people, especially those included in various forms of learning, may experience an exacerbated level of stress, anxiety, depression, or frustration, with undesirable consequences in terms of academic performance, social life, or even health [22]. Sources of stress for students include academic activity, personal situation, work environment, time, and economic circumstances [23].

There has been an increase in cases of bruxism among young people, especially in universities, with epidemiological studies indicating an increase from 5% in 1966 to 22% in 2002, with the level of stress following the same upward trend [24,25].

The aim of the present study was to establish the prevalence of sleep/awake bruxism among young students in Transylvania and to correlate the presence of this type of muscle activity with the behavioral variations listed above. The study hypothesis is represented by the increase in bruxism among students in higher education and the stress (job-related stress anxiety or depression) prevalence in the same population.

## 2. Materials and Methods

This analytical, observational, cohort, cross-sectional and prospective study involved 308 volunteer subjects of different nationalities, all students in “Iuliu Hațieganu” University of Medicine and Pharmacy Cluj-Napoca, Romania.

The study protocol complied with the Helsinki Convention and received the approval with no. 184/19.04.2018 from the Ethics Commission of the University of Medicine and Pharmacy “Iuliu Hațieganu”, Cluj-Napoca.

The study design and the questionnaire were presented to the students at the end of the prosthetic course (study years 4, 5, and 6), and participation was voluntary. The inclusion criteria in the study were clinically healthy students, with or without bruxism. The exclusion criteria from the study were the presence of an associated general pathology, chronic treatment with anti-inflammatories, antidepressants, anxiolytics, or agonists or antagonists of dopamine or serotonin, ongoing orthodontic treatments, ongoing dental treatments, or the presence of exhaustive prosthetic rehabilitation (fixed prostheses with more than three units), which may interfere with the functional movements of the mandible.

Subjects included in the study were informed about the structure of the study and the objectives pursued, and they signed an informed consent form before starting the research. The study was conducted in the Department of Dental Prosthetics of the Faculty of Dentistry, between June and October 2018, during the semester, excluding the exam period. The questionnaires were distributed during the breaks and were completed by participants within 10 min. The questionnaire was anonymous (filling in the name and surname was optional) and was structured in five sections. For the Romanian subjects, the questionnaire was in Romanian; for the other participants in the study, an English version was available.

The first section included demographic data related to age, sex, nationality, background, and level of education.

The second section contained five questions related to the presence of bruxism and occlusal status. This section was based on the Fonseca Questionnaire, a tool used to assess the presence and severity of temporomandibular disorder (TMD).

Question 8 (“Do you grit your teeth?”) was divided into two questions (“Do you grit your teeth during the day, when you are awake?” and “Do you grit your teeth during the night/sleep?”) to identify the type of bruxism: sleep bruxism or awake bruxism. We added additional questions about the presence of pain in the temporomandibular joint (TMJ), muscle pain or fatigue in the morning, on waking, the degree of satisfaction with their own occlusion (given that all subjects had a level of knowledge that allowed a relevant assessment), and consumption of analgesics and anti-inflammatory drugs. Answers were scored with values from 1 to 5, where 1 = never and 5 = always.

The subsequent sections aimed to identify and quantify psychological disorders. Before each set of questions, the following clarifications were made, in order not to orient the answers, altering their validity: “There are no correct or wrong answers. Do not waste too much time on any statement!”

The third section featured the DASS-21 questionnaire, a test comprising three sets of statements about the emotional status of depression, anxiety, and stress in general, in the subject. DASS-21 is based on quantifiable aspects of psychological disorder, rather than on their qualitative assessment [26]. The design of DASS-21 (which is, in fact, the reduced version of the DASS-42) is based on the observation (also confirmed by researchers) [27,28] that the differences among depression, anxiety, and stress in normal subjects compared to those with a pathological component are a few degrees (Table 1). The answers were scored between 0 = this statement does not apply to me at all and 3 = this statement applies to me a lot/most of the time [29]. The score for depression, anxiety, and stress was calculated by summing all the answers and comparing them with the evaluation scale of the questionnaire, published by the authors:

The fourth section was based on the test of irritation, anxiety, and depression conditioned by duty (pregnancy, job), developed by Caplan, Cobb, French, Van Harrison, and Pinneau (1980) [30]. In the present study, the duty of our subjects, who were students, was considered the learning process in the context of the school curriculum. This test, which includes 13 statements, measures three dimensions of behavioral disorders: depression, anxiety, and related irritation, generated by duty. Answers were scored on a scale from 0 to 4, where 0 = never and 4 = most of the time [30].

The last section of the questionnaire analyzed the state of frustration experienced by the patient in direct relation to their duty (task)—in our case, that of learning. This was based on the questionnaire “frustration and work” developed by Peters, O’Connor, and Rudolf (1980), which includes three statements that follow the level at which the subject finds their job frustrating. The answers were obtained on a scale from 1 to 7, where 1 = do not agree at all, and 7 = strongly agree [31]. To certify the correctness of the answers, the third question had the opposite answer. The interpretation of the frustration score is as follows: a total value ≤10 is within normal limits, values between 11 and 15 indicate an average level of frustration, and a score >15 is associated with frustration, as a psychosocial syndrome.

The data obtained after completing the questionnaires were transferred in electronic format, to the Microsoft Excel program, obtaining the database necessary for the final analysis. Statistical analysis was performed using MedCalc Statistical Software version 19.1 (MedCalc Software bvba, Ostend, Belgium; http://www.medcalc.org (accessed on 4 May 2019). Continuous data were analyzed for normal distribution with the Shapiro–Wilk test and were expressed as medians and 25th and 75th percentiles. Qualitative data were expressed as frequencies and percentages. The Mann–Whitney test was used for comparisons between groups. The Spearman rho coefficient was used to examine the correlation between continuous variables. The level of statistical significance was set at *p* < 0.05.

## 3. Results

### 3.1. Demographic Data

This study involved 308 young subjects of both sexes, of different nationalities, aged between 19 and 30 years old, with an average age of 23.82 years.

The demographic data of the subjects are presented in the Table 2.

### 3.2. Occlusal Status Assessment

The questionnaire focused on joint and occlusal status revealed a mean score of 3.67 subjects for the degree of satisfaction with their own occlusion, higher for men (score = 3.79) than for women (score = 3.57), while the level of muscle fatigue and pain in the morning when waking up was rated with an average value of 1.85 (1.69 for men and 1.97 for women). The use of analgesics was higher among women (score = 1.58) than among men (score = 1.36) (Figure 1).

### 3.3. General Psychological Status Evaluation

The analysis of the psychological status of the subjects included in this study revealed general and comparative data regarding the two sexes (Table 3).

### 3.4. The Correlation between Sleep/Awake Bruxism and Psychological Status

Of the 308 participants in the study, 31 (10.06%) subjects presented both AB and SB, 16 (5.19%) presented only AB, and 70 (22.7%) presented only SB (Figure 2).

The results obtained from the questionnaires were analyzed separately for sleep bruxism and for awake bruxism.

The 47 subjects with AB had an average age of 24.68 years, slightly increased compared to the group without AB, and the distribution was approximately equal for men and women (24 women and 23 men). The average score obtained by analyzing the answers can be found in Table 4.

From the total number of subjects with AB, five presented a pathological score for frustration. The results regarding psychological aspects (anxiety, depression, and stress) were significantly increased compared to the control group. The appreciation of one’s own occlusion was slightly unsatisfactory among those with AB, without major differences. In contrast, muscle fatigue was more pronounced in daytime bruxists.

For the 101 nocturnal bruxists, the average age was 23.84, also with an equal distribution between the two sexes (51 women vs. 50 men). A summary of the responses regarding occlusion and psychological status can be found in Table 5.

Data obtained from subjects with SB indicated a higher score, denoting psychological involvement compared to the group without bruxism, but lower than that recorded in the group with AB. Satisfaction with one’s own occlusion was lower in those with SB than in those with AB, with a significant difference (Figure 3).

The statistical analysis of the results aimed to establish correlations between SB and AB and epidemiological factors (age, sex) and psychological factors (anxiety, stress, depression, frustration) or their conditioning in relation to duty (that of learning). We did not find any statistically significant correlation between AB or SB and age (*p* = 0.30 and *p* = 0.37, respectively), sex (*p* = 0.44 and *p* = 0.48, respectively), or nationality (*p* = 0.55 and *p* = 0.67, respectively). Only a high degree of stress and frustration (*p* = 0.035 and *p* = 0.020) was observed in European subjects, except for the Romanians and the French, likely related to the difficulties of adapting to the language and lifestyle in Romania. Female subjects were statistically significantly associated with an increased level of stress (*p* = 0.004), duty-related depression (*p* = 0.006), and duty-related anxiety (*p* = 0.003).

Both SB and AB were statistically significantly associated with muscle fatigue and masticatory pain in the morning when waking up (*p* = 0.00).

The correlations of AB and SB with psychological factors are highlighted in Table 6 and Table 7.

## 4. Discussion

The everyday ongoing busy schedule, daily responsibilities, geo-political changes, and COVID-19 pandemic are elements that generate stress and anxiety. The onset of bruxism in correlation with psycho-emotional factors can be considered a public health problem, because, at a certain point in life, each individual can present this pathology, more or less consciously. It is advisable to inform the population about this muscular activity, in order to be able to limit the negative effects on the dento-maxillary apparatus.

In the present study, we analyzed the prevalence of bruxism and the associations between the two types of bruxism and general psychological status, as well as the duty-related psychological reactions (depression, anxiety, stress, and frustration related to learning—as a task).

The data collected from the literature on the prevalence of bruxism are contradictory, depending on the group studied. In our case (the subjects were young adults), we found a prevalence of 15.2% for AB and 32% for SB, values higher than those found by Wetselaar et al. in the Danish population, at the same age (6.6% for AB and 20% for SB) [7]. Serra-Negra et al., studying the quality of sleep and the prevalence of bruxism among students at the Faculty of Dentistry in Brazil, found values of 36.5% for AB and 21.5% for SB [32]. The differences obtained can be explained by the age of the subjects included in the study; in our case, the subjects were between 19 and 30 years old, while the Brazilian students were between 17 and 47 years old, proving that the prevalence of bruxism decreases with age [33]. Molina et al. indicates an increased incidence of both types of bruxism in females, associated with an increased level of stress and anxiety [34]. Cavalo, in 2016, in a study with student subjects, obtained a higher prevalence of SB (31.8%) close to that obtained in our study, without differences between the two sexes [35]. Our results support the increased involvement of psychological factors in females, with no gender differences in the presence of bruxism. Nationality may be a factor associated with the presence of bruxism, but more in relation to the level of stress generated by the adaptation of foreigners to different cultures and values, such as, in our study, European students, other than Romanian or French (who are closer in language and culture).

The gnashing or collision of teeth during the day or during the night is considered to have a major impact on the etiopathogenesis of TMD [36]. The dimension of the effect and how daytime activities interfere with nighttime activities are not fully clarified. What is certain is that TMD pain is associated with both AB and SB, according to data published by Sierwald et al., in a study that included 733 patients who reported at least one painful episode in the dento-maxillary apparatus [37]. The diagnosis of self-reported bruxism is made when the subject complains of, in addition to the activity of gnashing/collision of teeth, muscle fatigue and/or pain in the dento-maxillary apparatus or the presence of localized or generalized dental wear surfaces [1]. In the subjects included in this study, the score on the presence of pain or muscle fatigue was equal for SB and AB (score = 2.63) and higher than the average score of subjects without bruxism (score = 1.85). The correlation between AB/SB and muscle fatigue or pain at the level of the dento-maxillary apparatus had statistically significant values (*p* = 0.00).

The evidence-based scientific data, which support the multifactorial etiology of bruxism, genetic influence, and psychological factor involvement, have increased [38]. Personality characteristics, such as sensitivity to stress and anxiety, are the main psychological factors associated with the presence of bruxism in both children and adults [39]. The pathophysiological mechanism via which stress influences the presence of bruxism is explained by the fact that individuals with an increased level of neuroticism and anxious expectations tend to release emotional tensions by engaging in these bruxism activities [38].

A study published in 2017 by Ella et al., in subjects with bruxism, with or without craniofacial dystonia, found a statistically significant association between bruxism and stress (*p* < 0.001), but without identifying the bruxism type [40]. Cavallo described the same association between stress and bruxism, with statistical significance only in men [35]. Other studies highlighted the involvement of stress in the onset of bruxism [38,41], especially for the diurnal type. On the other hand, there are studies that refuted this association, considering it purely accidental [11]. Our research separately evaluated AB and SB, and the results revealed a statistically significant association with stress in both groups (*p* < 0.001 and *p* = 0.03).

Psychological factors have been recently associated with the chronotype of each subject, i.e., with the period of the day (24 h) in which maximum activity is reached. Compared to the chronotype, there are morning active subjects and evening active subjects. Taking into account the bruxism–psychological disorders–chronotype relationship, certain duties/activities that come outside one’s chronotype can cause an increase in stress level, expressed by the appearance of bruxism [42].

Along with stress, depression and anxiety have been mentioned in the etiopathology of bruxism, but the association between bruxism and depression has not reached statistically significant values in the studies published so far [43]. For anxiety, Montero et al. found a statistically significant association with SB [44]. The results of our study indicated a statistically significant association among depression, anxiety, stress, and AB. SB was only associated with anxiety and stress.

The novelty of this research is the introduction in the study of specific aspects of psychological disorders such as depression, anxiety, and stress related to a task/duty of the subject (learning for our subjects). The aspects mentioned above appeared only in relation to the respective duty, with the subject not having a general psychological diagnosis. The results indicated a statistically significant association between AB and all three duty-related disorders, whereas SB was associated only with duty-related anxiety and duty-related stress.

Although, in the study group, the subjects had high values of the frustration index, the association between any form of bruxism and frustration was not statistically supported.

The limitations of the study were as follows: all the subjects included in the study were students in the same university, the bruxism was self-reported, and the method of collecting data was only survey-based. Only the presence of ‘possible bruxism’ could be detected, which might have led to a significantly higher prevalence reported than in reality.

As a recommendation, bruxism awareness among the general population should be included as part of the national public health education.

## 5. Conclusions

The present research indicated a statistically proven involvement of psychological factors in the occurrence of bruxism, both sleep and awake. Stress and anxiety can be favorable factors in the appearance of both types of bruxism; however, depression was associated only with awake bruxism. The study showed an association between psychological disorders related to duty and the presence of bruxism, an aspect not yet introduced in the literature. Frustration is also an uninvestigated psychological disorder in relation to bruxism, but the bruxism–frustration association was not statistically supported in the present study.

## Figures and Tables

**Figure 1 ijerph-19-00691-f001:**
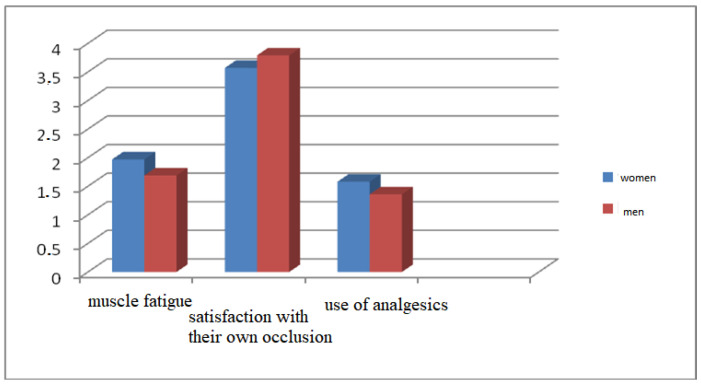
Occlusal status assessment.

**Figure 2 ijerph-19-00691-f002:**
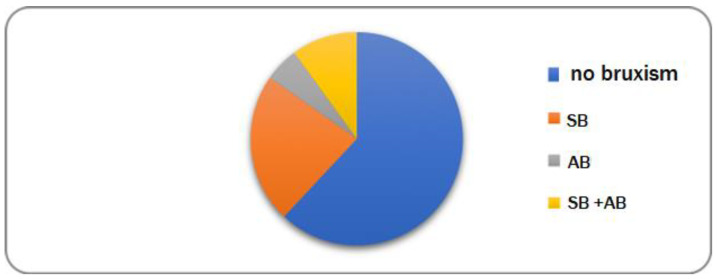
Frequency of bruxism in the studied group.

**Figure 3 ijerph-19-00691-f003:**
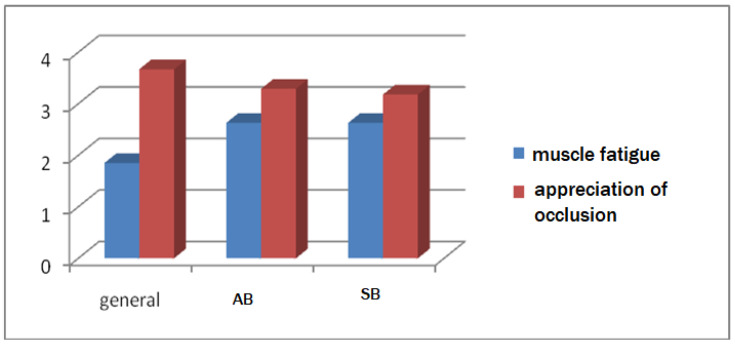
Comparison between AB and SB regarding muscle fatigue and occlusion assessment.

**Table 1 ijerph-19-00691-t001:** DASS evaluation scale (after Lovibond PF et al. [28]).

Severity	DASS Depression	DASS Anxiety	DASS Stress
Normal	0–9	0–7	0–14
Low	10–13	8–9	15–18
Moderate	14–20	10–14	19–25
Severe	21–27	15–19	26–33
Extremely severe	>28	>20	>34

**Table 2 ijerph-19-00691-t002:** Demographic data.

Parameters		No. of Subjects
Age (years)		23.82
Sex	M	133 (43.18%)
	F	175 (56.81%)
Nationality	Romanians	151 (49.02%)
	French	94 (30.52%)
	Other europeans (Germans, Italians)	45 (14.61%)
	Arabs	18 (5.84%)

**Table 3 ijerph-19-00691-t003:** The general psychological status of the studied group.

Psychological Disorders	Overall Score	Male Score	Female Score
Depression	3.29	3.11	3.42
Anxiety	3.97	3.57	4.26
Stress	5.63	4.89	6.20
Job/duty-related stress	6.35	5.37	7.09
Job/duty-related anxiety	5.66	5.06	6.12
Job/duty-related stress	3.60	3.41	3.75
Frustration	8.7	8.68	8.8

**Table 4 ijerph-19-00691-t004:** Results obtained in subjects with AB compared to those without AB.

Psychological Disorders	Score for Subjects without AB	Scote for Subjects with AB
Satisfaction with own occlusion	3.74	3.29
Muscle fatigue/pain in the morning	1.71	2.63
Depression	2.95	5.19
Anxiety	3.71	5.40
Stress	5.28	7.60
Learning/duty-related depression	6.12	7.63
Job/duty-related anxiety	5.46	6.76
Job/duty-related stress	3.38	4.83
Frustration	8.66	9.29

**Table 5 ijerph-19-00691-t005:** Results obtained from subjects with SB, compared to those without SB subjects.

Psychological Disorders	Score for Subjects without SB	Score for Subjects with SB
Satisfaction with own occlusion	3.90	3.18
Muscle fatigue/pain in the morning	1.47	2.63
Depression	3.12	3.64
Anxiety	3.48	4.96
Stress	5.36	6.20
Learning/duty-related depression	6.19	6.68
Job/duty-related anxiety	5.48	6.03
Job/duty-related sterss	3.35	4.12
Frustration	8.66	8.93

**Table 6 ijerph-19-00691-t006:** The correlation between awake bruxism and psychological status.

Parameters	Correlation Coefficient	*p*
Depression gen.	0.160	0.005
Anxiety gen.	0.143	0.012
Stress gen.	0.203	0.000
Learning-related depression	0.151	0.008
Learning-related anxiety	0.167	0.003
Learning-related stress	0.198	0.000
Frustration	0.076	0.184

**Table 7 ijerph-19-00691-t007:** The correlation between sleep bruxism and psychological status.

Parameters	Correlation Coefficient	*p*
Depression gen.	0.061	0.288
Anxiety gen.	0.193	0.001
Stress gen.	0.124	0.030
Learning-related depression	0.084	0.140
Learning-related anxiety	0.114	0.046
Learning-related stress	0.141	0.013
Frustration	0.107	0.060

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
