# Peer review of "The Epidemiology of Bruxism in Relation to Psychological Factors"

_ijerph, 2022, doi:10.3390/ijerph19020691_

Round 1

Reviewer 1 Report

Authors state that the aim of the present study was to establish the prevalence of sleep and awake bruxism among young students in Transylvania and to correlate the presence of bruxism with the behavioral variations. This study was an analytical, observational, cohort, cross-sectional and prospective and involved 308 volunteer subjects, of different nationalities, all students of the "Iuliu Hațieganu" University of Medicine and Pharmacy in Cluj-Napoca. Results are based only on a survey without using more objective methods of assessing bruxism. Although, the topic of the study seems to be quite interesting, I believe that many changes should be introduced to the text before publication of this manuscript.

Below, I present specific comments/questions which the Authors might want to take into consideration:

Abstract:
1. Abstract does not stand alone. It is not explanatory in the context of this study. Materials and methods, results, discussion and conclusion sections should be better described in the abstract.
2. The proper nomenclature in accordance to bruxism should be used. That is sleep bruxism and awake bruxism, bruxism described as muscle activity or behavior, etc. (this suggestion refers to all the text).
3. Which of the aspects connecting bruxism and psycho-emotional disturbances you mean as the one not yet presented in the literature? Please precise.

Introduction:
1. I believe that there is a better and more current definition of bruxism available. Authors can consider adding the latest definition of sleep bruxism that is "a masticatory muscle activity during sleep that is characterized as rhythmic (phasic) or non-rhythmic (tonic) and is not a movement disorder or a sleep disorder in otherwise healthy individuals" proposed in Lobbezoo et al. International consensus on the assessment of bruxism: Report of a work in progress. J Oral Rehabil. 2018 Nov;45(11):837-844. What is more researchers propose to use two separate definitions for sleep and awake bruxism. Please, use also these definitions.
2. Citation is needed for the sentence According to literature, the prevalence of bruxism does not vary according to nationality or race.
3. Citation is needed for this sentence The presence of bruxism can vary throughout life depending on the patient's age, with a higher frequency in young adults and lower in the elderly.
4. Citations are needed for this sentence Stress, anxiety and depression are the psychological factors most commonly associated with the presence of bruxism.
5. Citations are needed for the part of the text presented below:
In many cases, the pathophysiological complications of diseases occur as a result of stress, and subjects exposed to stress, those who work or live in a stressful environment, are more likely to develop various diseases. Psychological disorders can lead to pathological conditions, such as:
- physiopathological reactions: tachycardia and palpitations, migraines, vertigo, intestinal transit disorders, fatigue;
- cognitive reactions: memory disorders, attention deficit, decreased ability to concentrate, reduced flexibility;
- emotional reactions: irritability, repression of emotions, emotional instability, anxiety, depression;
- behavioral reactions: sleep disorders, inefficient time management, excessive consumption of alcohol and / or drugs, the development of hostile, aggressive behaviors.
Anxiety is a pathological affective state characterise by psychomotor anxiety, unexplained fear, without object, or related to the possibility of imminent danger or failure.
Anxiety is perceived by the subject as an emotional disorder translated into an indefinite feeling of insecurity. It describes a "normal" anxiety, which creates an optimal environment for learning and performance and a pathological anxiety, in which the subject is so deeply marked that he can no longer control himself. The pathological form is accompanied by multiple vegetative reactions. Depression manifests itself as a state of sadness, pessimism, general disinterest, lack of power, but also the desire to concentrate and communicate. It can have as a starting point a failure, a loss, a state of accentuated or prolonged stress, certain failures or incapacities. In moderate depression, the subject shows inhibition in expression, pessimistic attitude. In cases of deep depression, the facies is altered, expressionless, the subject lacks initiative.
6. Please describe the association between the co-occurrence of bruxism and psychoemotional status better in the light of existing literature to better show the rationale for conducting this kind of study.
7. I believe that there are more current and relevant studies to cite in the introduction section.
8. Bruxism is not considered as parafunction anymore. Please use the proper nomenclature (this suggestion refers to all the text).
9. What is the study hypothesis?

Materials and methods:
1. How the participants were recruited for the study?
2. Why were the ongoing orthodontic treatments, ongoing dental treatments or the presence of exhaustive prosthetic rehabilitation (fixed prostheses with more than 3 units), which may interfere with the functional movements of the mandible exclusion criteria? Please explain.
3. Please provide the exact questionnaires used and describe the rationale of using them.
4. As we already know that self-reporting data are insufficient to diagnose bruxism, why did not Authors use the more objective diagnostic methods? This is a serious limitation of this study. Please discuss it.

Results:
1. Results section should be divided into subsections, better organized and described more thoroughly.
2. The quality of tables is insufficient.

Discussion:
1. The discussion should be better organized.
2. The discussion lacks a description of the clinical utility of the results obtained in this study.
3. Taking into account the obtained results, can Authors indicate any recommendations for clinicians?
4. Citations are needed for the part of the text presented below:
The gnashing and collision of teeth during the day or during the night is considered to have a major impact on the etiopathogenesis of TMD. The dimension of the effect and how daytime activities interferes with nighttime activities are not fully clarified.
5. The fact that only survey-based methods were used to collect the data is serious limitation of presented study. The Authors should discuss all the limitations of the study.

Reviewer 2 Report

Manuscript is nicely written. However, there are a few suggestions. 

  • please base your definition of bruxism on the 2018 International Consensus (https://onlinelibrary.wiley.com/doi/full/10.1111/joor.12663)
  • bruxism is no longer considered a parafunction 
  • consider using "bruxist" instead of "bruxomaniac" 
  • who evaluated the psychological tests?

Round 2

Reviewer 1 Report

I would like to thank the Authors for improving the work and referring to my comments. However, I still have a few comments that the Authors might want to take into consideration to improve their manuscript.

Abstract:

1. Line 65: I suggest changing night/day bruxism into sleep/awake bruxism (the comment refers to all the text).

Introduction:

1. In accordance to 2018 International consensusregarding bruxi there are more specific definitions proposed for sleep and awake bruxism. I suggest using them.
Second, as sleep and awake bruxism are generally considered as different behaviours observed during sleep and wakefulness, respectively, the single definition for bruxism is recommended to be “retired” in favour of 2 separate definitions:
Sleep bruxism is a masticatory muscle activity during sleep that is characterised as rhythmic (phasic) or non-rhythmic (tonic) and is not a movement disorder or a sleep disorder in otherwise healthy individuals.
Awake bruxism is a masticatory muscle activity during wakeful-ness that is characterised by repetitive or sustained tooth contact and/or by bracing or thrusting of the mandible and is not a movement disorder in otherwise healthy individuals.

2. Unfortunately, the Authors did not refer to my previous comment Please describe the association between the co-occurrence of bruxism and psychoemotional status better in the light of existing literature to better show the rationale for conducting this kind of study. This makes the introduction insufficient to indicate the need for such a study.

3. Please include the study hypothesis in the introduction section.

Materials and methods:

1. In the response for my revision Authors state that: Ongoing orthodontic treatments, ongoing dental treatments or the presence of exhaustive prosthetic rehabilitation (fixed prostheses with more than 3 units) have been excluded in order to reduce the possibility of involvement of malocclusion in the appearance of bruxism. For ongoing orthodontic treatments the occlusion is not stable, this represents also a possibility of bruxism onset. In accordance to my best knowledge, the occlusal factors are no more considered as influencing the bruxism occurrence or intensity. Please verify this more thoroughly.

2. Thank you for providing the full questionnaire used in this study. Is that a validated questionnaire? Are all questions a part of other validated questionnaires? If not, discuss it as study limitation.

Discussion:

1. The citation are needed of this part of the text The everyday ongoing busy schedule, the daily responsibilities as well as the geopolitical changes and the COVID-19 pandemic are elements that generate stress and anxiety. The onset of bruxism in correlation with psycho-emotional factors can be considered a public health problem, because at a certain point in life, each individual can present this pathology, more or less consciously. It is advisable to inform the population about this muscular activity, in order to be able to limit the negative effects on the dento-maxillary apparatus.

References:

1. There is a problem with references numbering. Please check it.
